# Endocannabinoid System Receptors at the Hip and Stifle Joints of Middle-Aged Dogs: A Novel Target for the Therapeutic Use of *Cannabis sativa* Extract in Canine Arthropathies

**DOI:** 10.3390/ani13182833

**Published:** 2023-09-06

**Authors:** Rodrigo Zamith Cunha, Giulia Salamanca, Fanny Mille, Cecilia Delprete, Cecilia Franciosi, Giuliano Piva, Alessandro Gramenzi, Roberto Chiocchetti

**Affiliations:** 1Department of Veterinary Medical Sciences, University of Bologna, 40126 Bologna, Italy; rodrigo.zamithcunha2@unibo.it (R.Z.C.); giulia.salamanca2@unibo.it (G.S.); fanny.mille@studio.unibo.it (F.M.); 2Department of Veterinary Medicine, University of Teramo, 64100 Teramo, Italy; agramenzi@unite.it; 3Laboratory of Cellular Physiology, Department of Pharmacy and Biotechnology (FaBiT), University of Bologna, 40126 Bologna, Italy; cecilia.delprete2@unibo.it (C.D.); cecilia.franciosi2@studio.unibo.it (C.F.); 4Veterinary Clinic Dr. Giuliano Piva, 41049 Sassuolo, Italy; giulianopivavet@gmail.com

**Keywords:** cannabinoids, cannabinoid receptor type 1, cannabinoid receptor type 2, G protein-coupled receptor 55

## Abstract

**Simple Summary:**

The endocannabinoid system (ECS) plays a crucial role in various processes in animals, including pain, inflammation, and immune function. In this study, the presence and distribution of specific ECS receptors (CB1R, CB2R, and GPR55) in the joints of middle-aged dogs was investigated. By analysing the synovial tissues from the hip and the stifle joints, it was found that both CB2R and GPR55 were more prominently expressed by the synoviocytes as compared to CB1R. In addition, immune cells, such as macrophages and neutrophils, also exhibited some of these receptors. This intriguing finding suggested that the receptors in the ECS, particularly CB2R and GPR55, could be promising targets for therapeutic interventions, such as using *Cannabis sativa* extract, to address arthropathies in dogs.

**Abstract:**

The endocannabinoid system (ECS) has emerged as a potential therapeutic target in veterinary medicine due to its involvement in a wide range of physiological processes including pain, inflammation, immune function, and neurological function. Modulation of the ECS receptors has been shown to have anti-inflammatory, analgesic, and immunomodulatory effects in various animal models of disease, including dogs with osteoarthritis. The goal of this study was to identify and compare the cellular expression and distribution of cannabinoid receptor type 1 (CB1R) and type 2 (CB2R) and the cannabinoid-related G protein-coupled receptor 55 (GPR55) on the synovial cells of hip and stifle joints of seven dogs of different breeds without overt signs of osteoarthritis (OA). The synovial membranes of seven hips and seven stifle joints were harvested post mortem. The expression of the CB1R, CB2R, and GPR55 present in the synovial tissues was investigated using qualitative and quantitative immunofluorescence and Western blot (Wb) analysis. Synoviocytes of the stifle and hip joints expressed CB1R, CB2R, and GPR55 immunoreactivity (IR); no significant differences were observed for each different joint. Cannabinoid receptor 2- and GPR55-IR were also expressed by macrophages, neutrophils, and vascular cells. The ECS receptors were widely expressed by the synovial elements of dogs without overt signs of OA. It suggests that the ECS could be a target for the therapeutic use of *Cannabis sativa* extract in canine arthropathies.

## 1. Introduction

In recent years, the endocannabinoid system (ECS) has materialised as a possible therapeutic target in veterinary medicine owing to its complicity in a plethora of physiological processes including pain, inflammation, immune function, neurological function, and body homeostasis [1,2]. The ECS consists of the endogenous cannabinoids N-arachidonylethanolamine (anandamide; AEA) and 2-arachidonylglycerol (2-AG), receptors, and enzymes which regulate the synthesis and degradation of endocannabinoids. Cannabinoid receptors, namely, cannabinoid receptor 1 (CB1R) and cannabinoid receptor 2 (CB2R), are widely distributed throughout the body, including the central nervous system, immune cells, and other peripheral tissues [3,4,5]. In addition, cannabinoids (exogenous and endogenous) and other *Cannabis sativa* compounds (terpenes and flavonoids) interact with a diversity of other receptors such as the G protein-coupled receptor 55 (GPR55), G protein-coupled receptor 119 (GPR119), transient receptor potential vanilloid 1 (TRPV1), transient potential receptor vanilloid 4 (TPRV4), and peroximase proliferator receptor alpha (PPARɑ) and gamma (PPARɣ) [6,7,8].

The endogenous ligands of CB1R and CB2R are AEA and 2-AG, respectively, two endocannabinoids which are produced by the cells “on demand” and play a role in a wide range of physiological and pathophysiological processes [9].

Endocannabinoids and phytocannabinoids, binding to the cannabinoid receptors of the membranes of the neurons, can modulate and inhibit the hyperactivity of primary afferent fibres and decrease the release of neurotransmitters [10].

Both AEA and 2-AG were found in the synovial fluid of osteoarthritic joints of dogs and their contralateral nonaffected joints; as a matter in fact, an increase in 2-AG levels was noted in the stifle joints of arthritic knees when compared to the contralateral joints [11]; the higher concentrations of 2AG in the affected joints may indicate that it plays a role in the pathophysiological process of joint diseases.

Recent studies have shown that the activation of CB1R by minor phytocannabinoids exerts anti-arthritis activity in murine models, highlighting its potential for the treatment of chronic inflammatory diseases such as rheumatoid arthritis (RA) [12]. Similarly, CB2R pharmacological activation in a mouse model of osteoarthritis (OA) showed a protective effect, indicating the potential role of CB2R in the pathogenesis of the disease [13].

G protein-coupled receptor 55, which is currently thought of as the third cannabinoid receptor (CB3R) [14], is a member of the endocannabinoid receptor family; in fact, one of its endogenous ligands is the endocannabinoid neurotransmitter lysophosphatidylinositol [15]. G protein-coupled receptor 55 is implicated in various physiological processes, including bone metabolism and inflammation [16], the regulation of osteoclast and osteoblast functions [17], and the reduction of pain in joint inflammation, indicating a potential therapeutic role for GPR55 in the treatment of joint diseases such as osteoarthritis and RA [18,19]. While the exact role of GPR55 in joint health is still being elucidated, current evidence suggests that it is a promising target for the development of novel therapeutics for joint disorders.

Modulation of the ECS receptors has been shown to have anti-inflammatory, analgesic, and immunomodulatory effects in various animal models of disease, including dogs with OA [20,21,22,23]. As such, targeting the ECS might represent a promising approach for the development of safe and effective therapies for a range of veterinary conditions. The ECS plays a crucial role in maintaining joint health and bone metabolism by modulating the activity of immune cells and reducing inflammation in both tissues, with evidence suggesting that it can regulate bone formation and resorption [24,25]. The mRNA and the immunoreactivity for the CB1R, CB2R, and GPR55 have already been described in the synovial membrane of the metacarpophalangeal joints of horses [1].

Understanding and correctly managing musculoskeletal diseases and arthropathies in veterinary patients is of great importance, since articulations can be affected by a diversity of pathologies such as arthritis, osteoarthritis (OA), septic arthritis, synovitis, capsulitis, and ligament rupture [26]. Studies have shown that approximately 20% of dogs will develop some form of joint disease during their lifetime, with an increasing incidence in older dogs; it has been found that in dogs older than 7 years, about 80% of them showed radiographic evidence of osteoarthritis (OA) in at least one joint [27,28,29,30,31]. In addition, a study on a population of dogs aged 10 years or older found that more than 40% of them had radiographic evidence of hip dysplasia, a common joint disease in dogs [32,33]. Canine hip OA shares anatomical/pathological characteristics with developmental dysplasia of the hip in humans, and therefore, canines have been proposed as the best spontaneous animal model for joint problems [34,35].

These findings emphasised the elevated prevalence of joint diseases in dogs, especially in older dogs [36], and highlighted the importance not only of an early diagnosis, but also effective management in order to improve their quality of life. It is vital to understand the underlying mechanisms of homeostasis of an organism if one wants to develop new drugs and properly use substances already available to manage and treat joint disorders and support healthy joints, thereby delaying the development of arthropathies. Knowledge of the species-specific distribution of the ECS receptors, known by its regulatory function of tissue homeostasis, represents the basic pillar for later clinical trials and novel therapies for each veterinary category of patients.

While there are few and contradictory publications regarding the therapeutic use of cannabinoid molecules [11,21,22] in osteoarthritic dogs, to the best of the authors’ knowledge, no literature is available regarding the expression and cellular distribution of cannabinoid receptors at the levels of the joints of canines.

The author’s hypothesis is that, similar to other mammalian species, middle-aged dogs will present different cellular distribution of the cannabinoid receptors in the synovial membrane and its adjacent structures, depending on the role each receptor plays in maintaining organic homeostasis. Therefore, the objective of the present study was to identify the expression and distribution of the CB1R, CB2R, and GPR55 at the synovial cells of the hip and stifle joints of dogs without overt signs of OA using qualitative and quantitative immunofluorescence and qualitative Western Blot (Wb) analysis.

## 2. Materials and Methods

### 2.1. Animals

The inclusion criteria for the selection of the animals were: animals from 2–15 years of age, male or female of any breed, and no clinical history of joint-related lameness or OA related to the hip and stifle joints. Synovial membrane tissues from the hip and stifle joints were collected post mortem from 7 dogs (3 males and 4 females, from 2 to 15 years of age (10 ± 4 years; average ± St. Dev.). A dog died from postoperative complications and 6 were euthanised for humane reasons due to different diseases (not involving OA); the patient which suffered from osteosarcoma had pain related to the cancer; the tissues were collected after owner consent was obtained (Table 1).

### 2.2. Tissue Collection

The synovial membranes of the hip and the stifle joints were analysed for the presence of OA. Macroscopically, the synovial fluid appeared normal in all the patients, and no signs of ischemia, cartilage disease, or bone disease were observed. Samples (~1 cm × 0.5 cm) of the lateral portion of the synovial membrane and synovial capsule from the hip and the stifle joints were harvested using a scalpel post mortem within 1 h of death.

Regarding the immunofluorescence, the tissues were gently pinned on balsa wood with entomological pins (with the synovial membrane face-up), fixed for 24 h at 4 °C in 4% paraformaldehyde in phosphate buffer (0.1 M, pH 7.2), and were subsequently processed to obtain cryosections (14 µm thick) which were later processed for immunofluorescence, as previously described [1]. For the Wb analysis, the tissues were placed in sterilised Eppendorf tubes which were immersed in liquid nitrogen and then stored at −80 °C.

### 2.3. Immunofluorescence

Hydration of the cryosections in phosphate-buffered saline (PBS) was carried out for immunostaining. The sections were incubated in a solution of 20% normal donkey serum (Colorado Serum Co., Denver, CO, USA), 0.5% Triton X-100 (Sigma-Aldrich, Milan, Italy, Europe), and bovine serum albumin (1%) in PBS for 1 h at room temperature (RT) to block nonspecific binding. The cryosections were incubated overnight in a humid chamber at RT with a mixture of primary antibodies (Table 2) diluted in 1.8% NaCl in 0.01 M PBS of 0.1% sodium azide. Following a wash in PBS (3 × 10 min), the sections were incubated for 1 h at RT in a humid chamber which contained the secondary antibodies (Table 3) diluted in PBS. The cryosections were again washed in PBS (3 × 10 min) and were subsequently mounted in buffered glycerol at pH 8.6 with the fluorescent stain 4′,6-diamidino-2-phenylindole—DAPI (Santa Cruz Biotechnology, Santa Cruz, CA, USA), which strongly binds to the adenine–thymine-rich regions of DNA.

As the receptors studied could have been expressed by different cellular types of the synovial membrane (synoviocytes and immune/inflammatory cells), different primary antibodies were applied in order to identify the intimal fibroblast-like (FLSs) and macrophage-like (MLSs) synoviocytes, subintimal macrophages and neutrophils, and vascular endothelial cells.

To identify the FLS and MLS, the antibodies directed against the fibroblast marker Vimentin and the macrophage/dendritic cell marker ionized calcium binding adapter molecule 1 (IBA1) were used, respectively.

To identify neutrophils, the antibody against calprotectin (MAC387) was used [37,38].

The endothelial cells were identified using the anti-endothelial marker CD31 antibody [39].

### 2.4. Specificity of the Primary Antibodies

The provider of the anti-CB1R antibody, raised in rabbit against the human CB1R, predicted cross-reactivity with the mouse, rat, and dog antigens. The sequence of canine CB1 protein is homologous (98.3%) to the sequence of human CB1 protein (https://www.uniprot.org/, accessed on 30 June 2018) [40]. In addition, the same antibody has been tested by Wb analysis on dog intestinal tissues [4].

In total, 2 anti-CB2 receptor antibodies directed against human CB2R were used in this study. The sequence of canine CB2R is the same (98.3%) as that of the sequence of the human CB2 protein (https://www.uniprot.org/). Dog tissues had already been utilised to test the specificity of the mouse anti-CB2 antibody (sc-293188) [37]. Dog tissues had not been used to test the specificity of the rabbit anti-CB2R antibody (PA1-744) using Wb analysis; however, in the current study, a double-staining method was used to colocalise the rabbit anti-CB2R antibody with the mouse anti-CB2R antibody.

The antibody anti-GPR55 receptor was raised against a 17 amino acid synthetic peptide of human GPR55 receptor. The sequence of canine GPR55 protein is homologous (83.5%) to the sequence of human GPR55 protein (https://www.uniprot.org/). The antibody provider indicated more (94%) cross-reactivity of this antibody with the canine GPR55 protein. In addition, dog nervous tissues were utilised to test the specificity of this antibody using Wb analysis [39].

In the present study, the specificities of the anti-CB1R, CB2R, and GPR55 antibodies were also tested on canine synovial tissues using Wb analysis (see below).

The anti-IBA1 antibody, which should recognise microglia in the central nervous system and macrophages/dendritic cells in the peripheral tissues [41], was raised in goats and is used against porcine IBA1. The dog IBA1 molecule has a 91.2% identity with the porcine molecule (https://www.uniprot.org/).

To identify the neutrophils, an antibody anti-calretinin (clone MAC387) was used [37]. It has recently been shown that this antibody does not recognise macrophages in canine tissues; however, it recognises neutrophils rather exclusively [37,38].

The antibody directed against the endothelial marker CD31 had already been used in dog tissues [39]. The antibody against vimentin had already been tested on canine tissues using Wb analysis [42].

### 2.5. Specificity of the Secondary Antibodies

The specificity of the secondary antibodies was tested by applying them after omission of the primary antibodies. No stained cells or protein bands were detected after omitting the primary antibodies.

### 2.6. Quantitative Analysis

Quantitative analysis of the intensity of the expression of CB1R, CB2R, and GPR55 in the synovial intimal layer was carried out on 7 dogs. For each animal, and each receptor, 3 randomly selected images of the synovial membrane (50 µm-thick and 100 µm wide; 5000 µm^2^ area) were acquired at high magnification (×40) using the same exposure time for all the images. ImageJ software (Image J, version 1.52t, National Institutes of Health, Bethesda, MD, USA) was used to analyse the signal intensity of each image; standardised thresholds were calculated empirically for brightness and contrast and were then applied to all images. The Color Histogram (gMEAN or rMEAN) tool of the software ImageJ (Image J, version 1.52t, National Institutes of Health, Bethesda, MD, USA) was then used to obtain the signal intensity.

Quantitative analysis of the number of cell layers of the synovial membrane and the cell density was carried out on 3 randomly selected areas (50 µm-thick and 100 µm wide; 5000 µm^2^ area); they were acquired at high magnification (×40) on 3 randomly selected images of the synovial membrane of the hip and of the stifle joint for each animal using a DAPI signal to stain the cell nuclei.

### 2.7. Statistical Methods

For each receptor, the mean of the 3 values/case of signal intensity in the 7 dogs was evaluated and compared. Statistical analysis was carried out using GraphPad Prism software (version 8.3, La Jolla, CA, USA). The normality distribution of the data was assessed using the Shapiro–Wilk test.

Comparisons between groups (receptors) within the same joint were carried out using the one-way ANOVA Tukey’s multiple comparisons test. A *p*-value ≤ 0.05 was considered significant.

Comparisons between groups (receptors) within the same joint and the 2 different joints were carried out using two-way ANOVA Tukey’s multiple comparisons test. A *p*-value ≤ 0.05 was considered significant.

Comparisons between the mean of each receptor at the 2 different joints were carried out using the Wilcoxon test and the paired *t*-test. A *p*-value ≤ 0.05 was considered significant.

Comparisons between the numbers of cell layers of the different joints were carried out using the Wilcoxon Test (nonparametric) and the paired *t*-test (parametric). A *p*-value ≤ 0.05 was considered significant.

Comparisons between the cell density of the different joints were carried out using the Wilcoxon Test (nonparametric) and the Paired *t*-test (parametric). A *p*-value ≤ 0.05 was considered significant.

Correlations between the number of layers of the synovial membrane and the density of the cells in the hip joint and stifle joint were carried out using the Pearson correlation test. A *p*-value ≤ 0.05 was considered significant.

### 2.8. Fluorescence Microscopy

The preparations were examined using a Nikon Eclipse Ni microscope equipped with the appropriate filter cubes to differentiate the fluorochromes utilised for differentiating between Alexa 488 and Alexa 594 fluorescence. The filter was set at 10 for the Alexa 488 (450–490 nm excitation filter and 515–565 nm emission filter) and the filter was set at 00 for Alexa 594 (530–585 nm excitation filter and 615 nm emission filter).

A Nikon DS-Qi1Nc digital camera and NIS Elements software Version 4.20.01 BR (Nikon Instruments Europe BV, Amsterdam, The Netherlands) were used to record the images. The same fluorochrome label was used for the 3 receptors, allowing for quantification. Corel Photo Paint was used to slightly adjust the contrast and brightness, whereas Corel Draw (Corel Photo Paint and Corel Draw, Ottawa, ON, Canada) was used to prepare the figure panels.

### 2.9. Western Blot

Tissue samples (hip and stifle synovial membranes) were collected from 3 dogs, frozen in liquid nitrogen, and stored at −80 °C until sample processing. An amount of 50 mg of tissue was fractioned into small pieces and homogenised in 500 µL of RIPA buffer (50 mM TRIS-HCl, pH 7.4, 100 mM NaCl, 1 mM PMSF, 1 mM EDTA, 5 mM Iodacetamide 1% Triton X-100, 0.5% sodium dodecysulphate) supplemented with a protease inhibitor cocktail (Sigma-Aldrich, Co, St. Louis, MO, USA). The extract was sonicated for 10 min at 20 s intervals every 2 min and pelleted for 20 min at 14,000 rpm. Total protein content was determined by Bradford method. Proteins (10 µg) were separated by 10–12% SDS–polyacrylamide gel and transferred to a PVDF membrane. After transfer, the membrane was blocked by 5% milk powder in PBST (PBS 0.01 M, pH 7.4) with 0.05% Tween 20 (Sigma-Aldrich, St. Louis, MO, USA) for 1 h at room temperature (RT). The membranes were incubated with primary antibodies (rabbit anti-CB1R, Orb10430; mouse anti-CB2R, Santa Cruz #sc293188; rabbit anti-GPR55, NB11055498) overnight at 4 °C, diluted 1:1000 in PBST 0.1% containing 1% milk. The following day, the membranes were rinsed 3 times with PBST, each for 15 min, and IgG horseradish peroxidase-conjugated secondary antibodies anti-rabbit (1:5000, Santa Cruz) and anti-mouse (1:5000, Sigma) were employed for incubation in 1% milk powder in PBST for 2 h at RT. After washout of secondary-HRP binding antibody, membrane was incubated with chemiluminescence substrate and developed with the enhancing chemiluminescence detection system (Santa Cruz Biotechnology or Cyanagen–Westar ηC ultra 2.0). Blots were visualised with the ChemiDocTM (Bio-Rad) imaging system.

## 3. Results

### 3.1. Western Blot Analysis

To determine whether the canine synovial membrane expresses proteins for CB1R, CB2R, and GPR55, Western blot analysis was performed. The present results showed that the anti-CB1R antibody revealed a band of 70 kDa, while the anti-CB2R antibody revealed a band of 55 kDa (Figure 1). The authors have recently demonstrated that anti-CB1R and anti-CB2R in the canine small intestine recognised slightly different bands; however, it should be emphasised that the present results in the canine synovial membrane were aligned with previous reports regarding the detection of CB1R (molecular weight between 35 and ~70 kDa) and CB2R (molecular weight of ~35 and ~60 kDa) using Western blot analysis [43,44,45,46]. The anti-GPR55 antibody recognised a major band around 35 kDa and its dimer at 70 kDa, as previously described [4,39], in the canine gastrointestinal tract (Figure 1, right panel). Negative controls, in which the primary antibodies were not involved in the incubation with the membrane, did not show bands (left panel).

The expression of CB1R, CB2R, and GPR55 in the canine synovial membrane was also confirmed on the protein level.

### 3.2. Immunofluorescence

#### Vimentin and IBA1 Immunoreactivity

A subset of cells lining the synovial membrane, i.e., FLSs, displayed prominent cytoplasmic vimentin immunoreactivity (vimentin-IR) in the hip and stifle joints (Figure 2a–c). In both joints, the FLSs were characterised by fusiform rounded nuclei and elongated, slender cytoplasmic processes. However, in the stifle, these cytoplasmic processes extended vertically toward the joint cavity within the different layers of the FLSs whereas, in the hip, the processes primarily extended horizontally.

Macrophage-like synoviocytes and subintimal macrophages expressed IBA1-IR; MLSs showed rounded nuclei and abundant cytoplasm (Figure 2d–f).

### 3.3. CB1R Immunoreactivity

Faint CB1R-IR was detected by the cytoplasm of the hip and stifle FLSs and MLSs. The CB1R-IR was detectable in seven of seven dogs (100%) and at both joints of the same dog (Figure 3a–f). Colocalisation studies showed that synoviocytes at both joints coexpressed CB1R and CB2R (Figure 3g–i). In some sections in which a small fragment of articular cartilage was present, it was possible to observe that the chondrocytes expressed moderate CB1R-IR.

Cannabinoid receptor 1 immunoreactivity was not expressed by the neutrophils or the endothelial cells either in the hip or in the stifle joints.

### 3.4. CB2R Immunoreactivity

A double-staining method was used to colocalise the rabbit anti-CB2R antibody with the mouse anti-CB2R antibody; both anti-CB2R antibodies were colocalised in the same synoviocytes and blood vessel cells (Figure 4a–d).

Moderate-to-bright CB2R-IR was detected by the cytoplasm of the hip and the stifle synoviocytes in seven of seven dogs (100%) by using both the anti-CB2R antibodies (from mouse and rabbit) (Figure 4a–e). Colocalisation studies showed that CB2R-IR was expressed by vimentin immunoreactive FLSs and IBA1 immunoreactive MLSs in both joints (Figure 4e–l).

Cannabinoid 2 receptor immunoreactivity was moderately expressed by MAC387-positive neutrophils (Figure 4m–p), CD31-positive endothelial cells (Figure 4m–p), and unidentified immune/inflammatory cells.

### 3.5. GPR55 Immunoreactivity

Moderate-to-bright GPR55-IR was expressed by the cytoplasm of the hip and the stifle synoviocytes in seven of seven dogs (100%). Colocalisation studies showed that the vimentin immunoreactive FLSs and the IBA1 immunoreactive MLSs expressed GPR55-IR at both joints (Figure 5a–l).

G-coupled protein receptor 55 was also brightly expressed by MAC387-positive neutrophils (Figure 5m–p), CD31-positive endothelial cells (Appendix A), unidentified immune/inflammatory cells, and chondroblasts.

There were no differences regarding the cellular distribution of the CB1R, CB2R, and GPR55 immunofluorescence at the stifle and hip joint elements.

### 3.6. Quantitative and Comparative Analysis of CB1R, CB2R, and GPR55 Immunoreactivity by the Synoviocytes

Quantitative analysis showed that there was less expression of CB1R, in comparison with CB2R (~*p* values of 0.0014 and 0.0020 for the stifle joints and the hips, respectively) and GPR55-IR (~*p* values of 0.00002 and 0.0001 for the stifle joints and the hips, respectively), in both the hip and the stifle joints of dogs without overt signs of OA.

Quantitative analysis also showed that the expression of CB2R-IR was statistically greater when compared with that of CB1R-IR (~*p* values of 0.0014 and 0.0020 for the stifle joints and the hips, respectively), but not statistically different when compared with that of GPR55-IR (~*p* values of 0.5655 and 0.2444 for the stifle joints and the hips, respectively).

Analogously, the analysis of the expression of GPR55-IR showed that it was statistically greater when compared with CB1R (~*p* values of 0.0002 and 0.0001 for the stifle joints and the hips, respectively), but not statistically different when compared with CB2R (~*p* values of 0.5655 and 0.2444 for the stifle joints and the hips, respectively) (Figure 6).

The cellular expression of CB1R, CB2R, and GPR55 in the synoviocytes of the hip and the stifle joints of dogs was additionally analysed using other statistical tests which indicated that there was no significant difference in the cellular expression of CB1R, CB2R, and GPR55 between the stifle and hip joints of dogs (Figure 7).

The expression levels of CB1R, CB2R, and GPR55 in the synoviocytes of the stifle and the hip joints in dogs were analysed using two-way ANOVA. The significance level (alpha) was set at 0.05. The ANOVA results indicated that neither the interaction between the row factor (dogs) and the column factor (joints), nor the individual factors, had a significant effect on the expression levels of the receptors and was not statistically significant (*p* = 0.9015), suggesting that the differences in expression across the dogs were not influenced by the joint type.

In summary, the two-way ANOVA results suggested that there were no significant differences in the expression levels of CB1R, CB2R, and GPR55 between the stifle and the hip joints of dogs. The lack of significant interactions and individual effects of dogs and joints indicated that the variation in receptor expression was not dependent on these factors. However, it is important to note that the difference in mean expression between the hip and the stifle joints was small and not statistically significant.

## 4. Discussion

The discoveries regarding the ECS receptors evidenced their important regulatory role in organic homeostasis and their involvement in several pathophysiological processes; the clinical and scientific demand is currently growing with respect to how best to use them as a therapeutic target. For this, one needs to first identify the presence or not of the ECS components in the organ of the pathology of interest.

Currently, the definition of the ECS is expanding to include other cannabinoid-related receptors [47,48]. This is the case, for example, for the GPR55, TRPV1, and nuclear PPARα, all of which are currently considered to be possible cannabinoid receptors [4].

Cannabinoid receptors are widely expressed through different cellular types of the organism; their distribution will be different depending on the organ and cell-type of interest [9]. Within the canine species, the expression of the CB1R, CB2R, and GPR55 was shown at the central nervous system (CNS) and the peripheral nervous system (PNS) [39,49,50], at the skin [37], and at the gastrointestinal tract [4]. Furthermore, the authors’ group recently showed the expression of cannabinoid and cannabinoid-related receptors at the synoviocytes (FLSs and MLSs) of horses [1], in which the mRNA and the immunoreactivity of CB1R was found in the synoviocytes of some but not all the subjects. The findings of the present study showed an important species-related particularity. In the equine species, only 71% of the equines expressed the CB1R by the synovial cells [1]; on the contrary, in the canine species, 100% of the dogs expressed the CB1R by the synoviocytes.

Although the exact factors which will regulate the ECS tone expression of the receptors related to species are still unknown, knowing the difference will reflect directly on the therapeutic choice for each veterinary patient. Moreover, the immunohistochemical expression of CB1R has been shown at the synovial membrane of all cats both with healthy joints and with degenerative joint disease [51]—and its upregulation was directly corelated to the degree of severity of the disease. Comparatively, the expression of CB1R is upregulated at the joint level in horses with synovitis [52,53]. Pointing to the role of CB1R in keeping a healthy joint environment, as well as its involvement within the pathogenesis of joint inflammation and its potential as a therapeutic target, new information regarding the moderate expression of CB1R by the synoviocytes and chondroblasts of dogs suggests a potential role in modulating pain and inflammation in joint tissue within canine species.

Although the use of the most known agonist molecule of the CB1R (Δ9-Tetrahydrocannabinol—THC) is still controversial, science can no longer deny the evidence of its use as a therapy for many pathologies. In a recently published study, Lowin et al. [54] were able to show the biphasic effects of THC on synovial fibroblasts from human patients with rheumatoid arthritis (synovial fibroblasts of rheumatoid arthritis—RASF) and peripheral blood mononuclear cells (PBMC) from healthy donors; THC provides proinflammatory and anti-inflammatory effects on the RASF and the PBMC. The effectiveness of THC in treating inflammation pertaining to rheumatoid arthritis may vary depending on the activating stimulus and the THC concentration. Therefore, it is important to titrate THC dosage to determine the therapeutic window.

Other minor phytocannabinoids also seem to exert therapeutic effects by means of CB1R modulation. Palomares et al. [12] showed that Δ9-Tetrahydrocannabinolic acid (Δ9-THCA-A), the precursor of Δ9-THC, can act as an orthosteric CB1R agonist; in vivo, Δ9-THCA-A reduced arthritis in collagen-induced arthritic mice, preventing the infiltration of inflammatory cells, synovium hyperplasia, and cartilage damage. Furthermore, Δ9-THCA-A inhibited the expression of inflammatory and catabolic genes on stifle joints; Δ9-THCA-A exerts anti-arthritis activity through the CB1R pathways, highlighting its potential in the treatment of chronic inflammatory diseases such as RA.

In the current study, the FLSs showed moderate/bright CB1R immunostaining, and only a few MLSs showed weak CB1R-IR. The authors’ hypothesis was that the CB1R pathway was directly involved in maintaining the structural integrity and physical barrier of the synovial membrane as well as regulating the synthesis of the synovial fluid. Therefore, one can reduce the inflammatory and degenerative synovium response by means of CB1R modulation [12,54]. The acquisition of this piece of information will directly determine the clinician’s therapeutic choice and positively change the case outcome. Molecule agonists of the CB1R would provide benefits for patients suffering from inflammatory joint disease directly at the pathological site, slowing the disease evolution and supporting the maintenance of a healthy synovial environment. However, more in vitro species-specific studies and clinical trials are needed.

Cannabinoid receptor 2 has already been identified in the synovium cells of mice [55], rats, humans [56,57], and horses [1]. In the present study, the FLSs, the MLSs, and the macrophage/dendritic cell antigen-presenters exhibited an elevated expression of CB2R. Furthermore, it was found that the FLSs coexpressed CB1R and CB2R, while the dendritic cells expressing CB1R also expressed CB2R. This indicates that synovial cells expressing CB1R also expressed CB2R, whereas not all the CB2R-expressing synovial cells expressed the CB1R. Notably, this difference in expression was more pronounced among the MLSs, which reinforced the role of the CB2R, rather than the CB1R, in regulating immune response.

Inflammatory processes within the stifle joint can alter the composition of the cruciate ligaments [58], and patients with a cruciate ligament rupture will have a higher density of macrophages and MLSs at the joint infiltrate and synovium [59]. By means of the CB2R pathways, one can regulate macrophage signalling and proinflammatory cytokine release; thus, as a result of the strong immunolabeling of MLSs and dendritic cells for the CB2R at the synovium of dogs, one could postulate that modulating its activity could benefit patients suffering from inflammatory degenerative joint diseases, specifically immune mediated diseases. In addition to cannabinoids, other compounds of the cannabis plant can interact with the CB2R, such as terpenes and β-Caryophyllene [60].

G protein-coupled receptor 55 is a relatively new and poorly understood cannabinoid receptor. It has been identified in a variety of cell types including sensory neurons, inflammatory cells, and bone cells [17]. Recent studies have shown that GPR55 is expressed in the synovial cells of horses [1], T cells and neutrophils of dogs [37], and chondrocytes of humans [61] and may play a role in regulating inflammation and immune response in joint tissue. In vitro studies have shown that activation of GPR55 in synoviocytes can increase the production of proinflammatory cytokines, such as Interleukin-6 (IL-6) and IL-8, which are associated with the pathogenesis of RA [54].

Neutrophils are important immune cells which infiltrate the synovium during inflammation [62]. Healthy joints are not expected to have an elevated presence of neutrophils; in the current study, very few neutrophils, scattered in the subintima of the synovial membranes, expressed cytoplasmatic GPR55-IR. Neutrophils in the synovial fluid of human patients with RA have been shown to express GPR55; its activation can induce neutrophil chemotaxis which can contribute to joint inflammation and damage in RA [63].

In the present study, FLSs, MLSs, and neutrophils (and chondrocytes) demonstrated GPR55-IR; these findings suggested that GPR55 may play a role in regulating synovial inflammation and joint destruction in inflammatory degenerative joint disease. Cannabidiol, being a GPR55 antagonist, may play a role in reducing the secretion of proinflammatory cytokines and in immune and inflammatory cell migration.

Minor cannabinoids, such as cannabigerol (CBG), Δ9-Tetrahydrocannabivarin (THCV), and cannabidivarin (CBDV), have been found to interact with GPR55 [64]. Δ9-Tetrahydrocannabivarin, a partial agonist of GPR55, is capable of inhibiting the activity of the full agonist lysophosphatidylinositol (LPI); CBG has also been shown to weakly inhibit the LPI response in GPR55 assays [65]. Another study found that CBD and other GPR55 antagonists can inhibit bone resorption in vivo; additionally, GPR55 ligands affect osteoclast formation in vitro, suggesting a potential therapeutic role for CBD and minor cannabinoids in bone disorders [17,66].

Chondroblasts are cells which produce and maintain the extracellular matrix of cartilage. G protein-coupled receptor 55 expression has been detected in human chondrocytes [67], and studies have shown that the activation of GPR55 in chondrocytes can induce the production of matrix metalloproteinases (MMPs), which are enzymes that degrade the extracellular matrix and contribute to cartilage destruction in arthritis [17]. Therefore, targeting GPR55 in chondrocytes may represent a potential therapeutic approach for slowing down cartilage destruction in dogs with arthritis, thus enhancing the welfare of older dogs, those most affected by spontaneous OA, using a molecular antagonist such as CBD.

Although there was no macroscopic or microscopic evidence of OA development or a history of lameness in the animals included in the small sample size of the present study, it is crucial to note that the median age of the animals was 10 years. This represented an important number of canine patients which could potentially be undergoing early development of OA or be suffering from subclinical OA, as age-related involution in dogs involves the loss of muscle mass and changes in the connective tissue and articular cartilage [36,68]. By discovering the cellular expression pattern of the ECS receptors in the joints of animals in this age group, one could speculate that it could be used as a target to treat and prevent the development of arthropathies in patients with elevated risk for the development of the disease.

The fact that no significant difference was found in the cellular distribution and expression of CB1R, CB2R, and GPR55 between hip and stifle joints without overt signs of OA is important in understanding the patterns of the ECS in these joints and in the organism. Understanding the patterns of the ECS in joints of mostly aged dogs provides a foundation for exploring its potential therapeutic applications in arthropathy treatment. Modulating the ECS using cannabinoid-based therapies or other approaches may offer promising avenues to alleviate pain and reduce inflammation in affected joints.

By recognising the similarities in the expression of CB1R, CB2R, and GPR55 in both the hip and the stifle joints of individuals without overt signs of OA, researchers and veterinarians can focus on developing targeted interventions which harness the ECS to restore joint health. Furthermore, the virtually slight (not statistically significant) difference in the mean expression of each receptor between the hip and the stifle joints was due to the different structure of each joint, as the stifle synovial membrane was shown to be composed of more cell layers than that of the hip joint; therefore, there are more cells to be analysed in the same area.

The results of the CB1R, CB2R, and GPR55 codistribution and coexpression within different cell types at different joint environments suggested that these receptors played a role in regulating inflammation and immune response in joint tissue and points to the complexity of the ECS. Additional research is warranted to fully elucidate the specific roles and interactions of the ECS receptors in joint health and disease, enabling the development of more effective and tailored treatment strategies for arthropathies in dogs.

## 5. Conclusions

The discovery of cannabinoid receptors (CB1R, CB2R, GPR55) in the synovial tissue of middle-aged dogs provides compelling molecular evidence supporting the use of cannabinoids for treating and delaying joint diseases. This breakthrough suggests the potential for developing the therapeutic agonists/antagonists targeting these receptors. Understanding the cellular expression of CB1R, CB2R, and GPR55 allows us to comprehend the role of the ECS in modulating inflammation, pain, and immune responses in canine synovia. This knowledge opens avenues for novel interventions utilising the ECS to maintain and enhance joint health and well-being in dogs.

## Figures and Tables

**Figure 1 animals-13-02833-f001:**
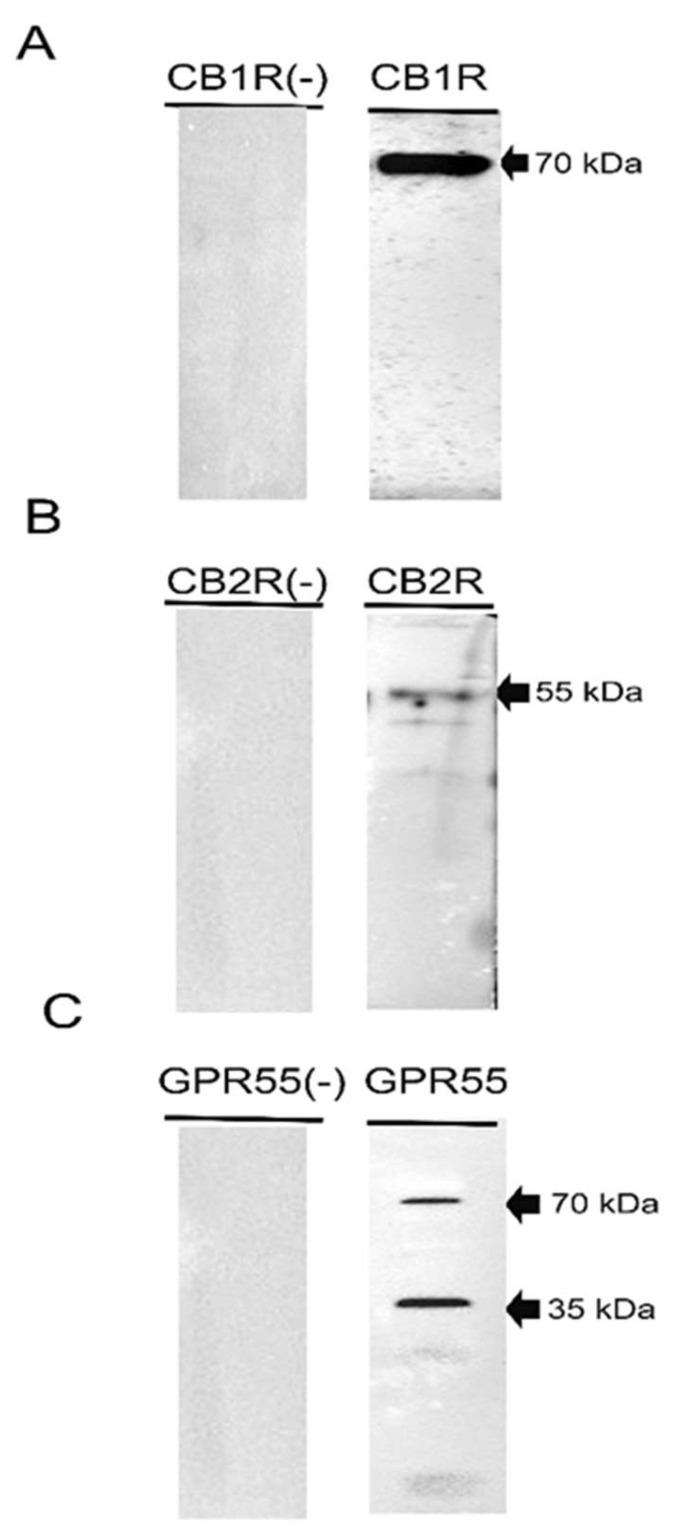
Representative image of Western blot analysis showing the specificity of the primary antibodies utilised (right panel): rabbit anti-cannabinoid receptor 1 (**A**), mouse anti-cannabinoid receptor 2 (**B**), and rabbit anti-G protein-coupled receptor 55 (GPR55) (**C**). Negative controls, in which the primary antibodies were not involved in the incubation with the membrane, did not show bands (left panel).

**Figure 2 animals-13-02833-f002:**
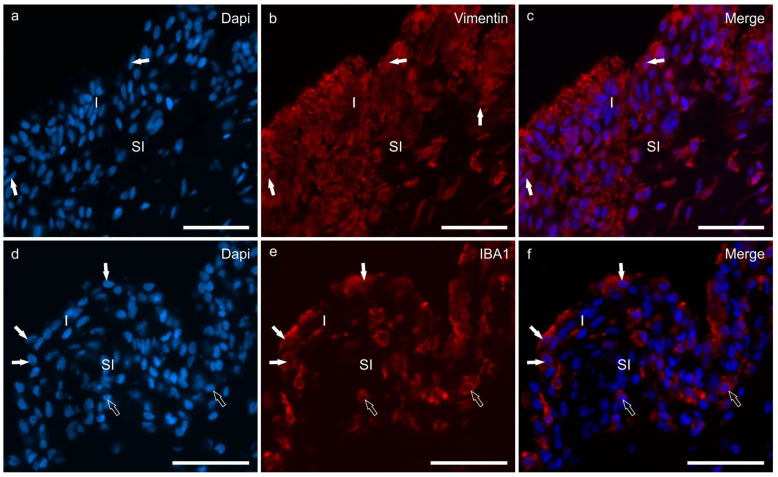
Photomicrographs of cryosections of the synovial membrane of the stifle joints of dogs showing immunoreactivity for the fibroblast marker vimentin (**b**) and the macrophage marker IBA1 (**e**). (**a**–**c**) The synovial membrane of the stifle joint showed different layers of synoviocytes (arrows) which expressed moderate-to-bright vimentin immunoreactivity (**b**). (**d**–**f**) Three macrophage-like synoviocytes lining the joint cavity, expressing bright IBA1 immunoreactivity, are indicated by the white arrows (**e**). The subintimal macrophages (open arrows) also expressed IBA1 immunoreactivity. Abbreviations: I, intima; SI, subintima. Bar: 50 µm.

**Figure 3 animals-13-02833-f003:**
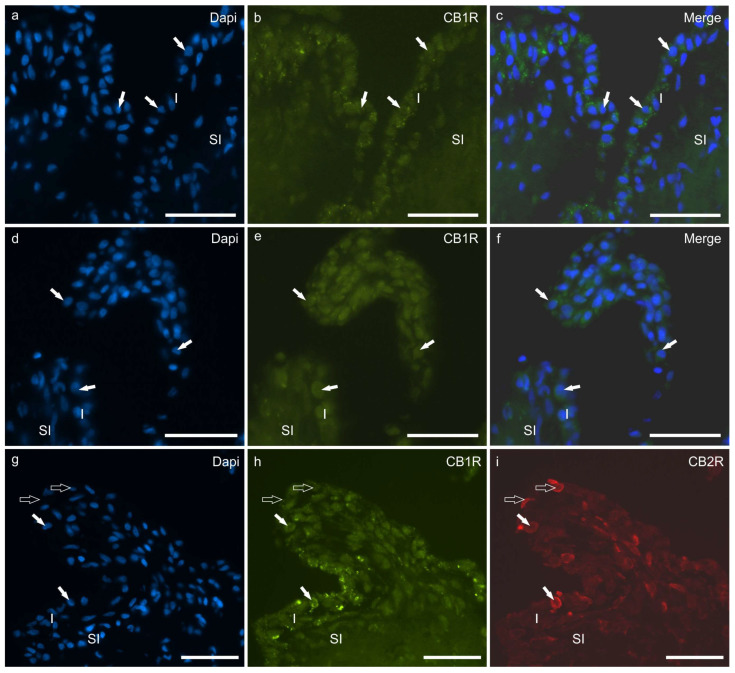
(**a**–**f**) Photomicrographs of cryosections of the synovial membrane of the hip (**a**–**c**) and stifle (**d**–**f**) joints of dogs showing immunoreactivity for the cannabinoid receptor type 1 (CB1R). The arrows indicate some synoviocytes showing faint CB1R immunoreactivity. (**g**–**i**) Photomicrographs of cryosections of the synovial membrane of the stifle joint of dog showing colocalisation between two antibodies directed against the cannabinoid receptor type 1 (CB1R) (**h**) and the cannabinoid receptor type 2 (CB2R) (**i**). The white arrows indicate two synoviocytes coexpressing CB1R and CB2R immunoreactivity. The open arrows indicate two synoviocytes which were immunoreactive only for the CB2R. Abbreviations: I, intima; SI, subintima. Bar: 50 µm.

**Figure 4 animals-13-02833-f004:**
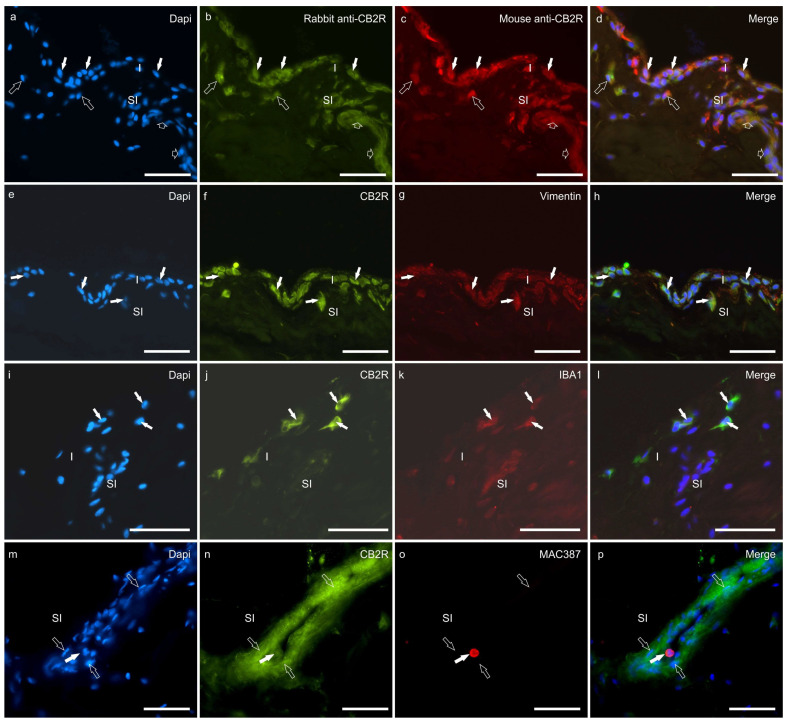
(**a**–**d**) Photomicrographs of cryosections of the hip synovial membrane of a dog showing colocalisation between two different antibodies directed against cannabinoid receptor 2 (CB2R). The white arrows indicate synoviocytes identified with the mouse anti-CB2R (**b**) and the rabbit anti-CB2R (**c**). The open arrows indicate two subintimal cells (likely inflammatory/immune cells) identified with both the anti-CB2R antibodies. The small open arrows indicate a subintimal capillary showing CB2R immunoreactivity. (**e**–**l**) Photomicrographs of cryosections of the synovial membrane of the hip (**e**–**h**) and stifle (**i**–**l**) joints of dogs showing immunoreactivity for the fibroblast marker vimentin (**e**–**h**) and the macrophage marker IBA1 (**i**–**l**). (**e**–**h**). The arrows indicate synoviocytes which showed bright cannabinoid receptor type 2 (CB2R) (**f**) immunoreactivity and moderate vimentin immunoreactivity (**g**). (**i**–**l**) The arrows indicate three macrophage-like synoviocytes which coexpressed bright CB2R immunoreactivity (**j**) and moderate IBA1 immunoreactivity (**k**). (**m**–**p**) Photomicrographs of cryosections of the synovial membrane of the stifle joint of a dog showing immunoreactivity for the cannabinoid receptor type 2 (CB2R) (**n**) and the neutrophils marker MAC387 (**o**). The open arrows indicate some endothelial cells of a subintimal blood vessel expressing bright CB2R immunoreactivity. The white arrow indicates one neutrophil inside the blood vessel coexpressing bright MAC387 and moderate CB2R immunoreactivity. Abbreviations: I, intima; SI, subintima. Bar: 50 µm.

**Figure 5 animals-13-02833-f005:**
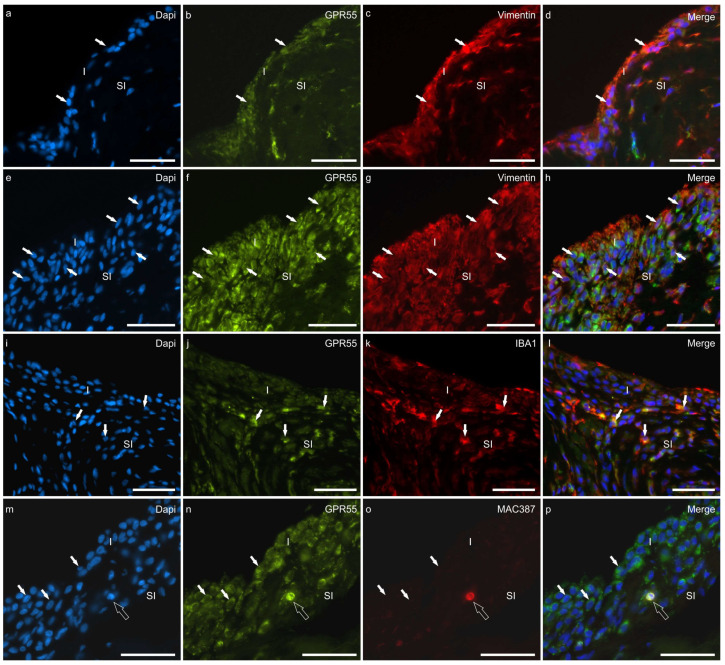
Photomicrographs of cryosections of the synovial membrane of the hip (**a**–**d**) and stifle (**e**–**p**) joints of dogs showing colocalisations of the antibody anti-G protein-coupled receptor 55 (GPR55) with the antibodies directed against the fibroblast marker vimentin (**a**–**h**), the macrophage marker IBA1 (**i**–**l**), and the neutrophils marker MAC387 (**m**–**p**). (**a**–**h**) the arrows indicate some synoviocytes (fibroblast-like synoviocytes) which were immunoreactive for GPR55 and vimentin. The different thicknesses of the epithelium lining the joint cavity of the hip (**a**–**d**) and stifle (**e**–**h**) joints (stifle > hip) should be noted. (**i**–**l**) The arrows indicate some macrophage-like synoviocytes which coexpressed GPR55 and IBA1 immunoreactivity. Given that the cut of the synovial membrane does not appear perfectly orthogonal, it cannot, however, be excluded that some IBA1 immunoreactive R cells are subintimal macrophages. (**m**–**p**) The white arrows indicate some synoviocytes expressing GPR55 immunoreactivity. The open arrow indicates one subintimal neutrophil, with a dapi-labelled polilobated nuclei, coexpressing GPR55 and MAC387 immunoreactivity. Abbreviations: I, intima; SI, subintima. Bar: 50 µm.

**Figure 6 animals-13-02833-f006:**
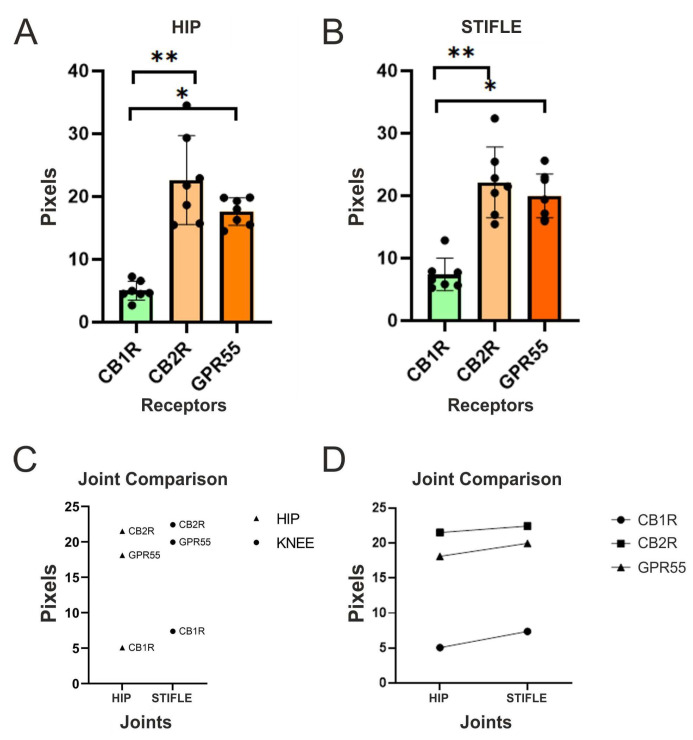
(**A**,**B**) Quantitative analysis of the hip (**A**) and stifle joint (**B**) using one-way ANOVA showed that the expression of CB2R-IR (**) was statistically different when compared with that of CB1R-IR, but not when compared with that of GPR55-IR. Analogously, the analysis of the expression of GPR55-IR showed that it is statistically different when compared with CB1R (*), but not with CB2R, at both joint sites. (**C**) To additionally examine the differences between the receptors, a Tukey’s multiple comparisons test was carried out. The results showed that the mean difference in expression between CB1R and CB2R was significant (mean diff = −16.19, *p* < 0.0001). Similarly, the mean difference between CB1R and the GPR55 was also significant (mean diff = −12.58, *p* < 0.0001). However, there was no significant difference in expression between CB2R and GPR55 (mean diff = 3.603, *p* = 0.1413, at both joints). (**D**) The statistical analysis carried out showed that there was no significant difference between the receptor expression of the different joints. In both the hip and the stifle joints, the receptors followed the same pattern of cellular distribution and expression (scatter dot plot with mean and SD).

**Figure 7 animals-13-02833-f007:**
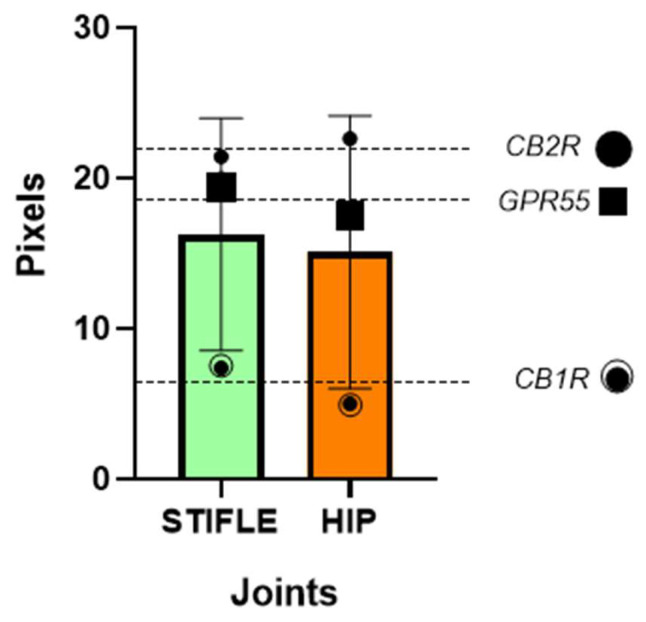
The Wilcoxon signed-rank test was carried out to assess the differences in expression between the mean of hip and stifle joints mean of each receptor. The Wilcoxon test results showed that the differences between joint were not statistically significant, with *p*-values of 0.2500 for both receptors (scatter dot plot with mean and SD).

**Table 1 animals-13-02833-t001:** Animals.

Dogs	Breed	Sex	Age (Years)	Cause of Death
#1	Golden retriever	Male, not neutered	9	Euthanasia/osteosarcoma left tibia
#2	Mixed breed	Male, not neutered	15	Euthanasia/mastocytoma
#3	Golden retriever	Female, spayed	9	Euthanasia/leukaemia
#4	English setter	Female, spayed	12	Euthanasia/mastocytoma
#5	Mixed breed	Female, spayed	2	Emergency/amputation of the right hind leg
#6	Mixed breed	Female, spayed	8	Euthanasia
#7	Pit bull terrier	Male, neutered	14	Euthanasia

**Table 2 animals-13-02833-t002:** Primary antibodies used in the study.

Primary Antibody	Host	Code	Dilution	Source
CB1R	Rabbit	Orb10430	1:200	Byorbit
CB2R	Mouse	sc-293188	1:50	Santa Cruz
CB2R	Rabbit	PA1-744	1:250	Thermo Fisher
Calprotectin	Mouse	M0747 Clone MAC387	1:400	Dako
CD31	Mouse	M0823 Clone JC70A	1:30	Dako
GPR55	Rabbit	NB110-55498	1:200	Novus Biol.
IBA1	Goat	NB100-1028	1:80	Novus Biol.
Vimentin	Mouse	IS630 Clone V9	1:600	Dako

Primary antibody suppliers: Alomone, Jerusalem, Israel; Dako Cytomation, Golstrup, Denmark; Biorbyt Ltd., Cambridge, UK; Novus Biologicals, Littleton, CO, USA; Santa Cruz Biotechnology, Dallas, TX, USA; Thermo Fisher Scientific, Waltham, MA, USA.

**Table 3 animals-13-02833-t003:** Secondary antibodies used in the study.

Secondary Antibody	Host	Code	Dilution	Source
Anti-goat 594	Donkey	ab150132	1:500	Abcam
Anti-mouse 594	Donkey	A-21203	1:500	Thermo Fisher
Anti-rabbit 488	Donkey	A-21206	1:1000	Thermo Fisher
Anti-rat 594	Donkey	A-21209	1:500	Thermo Fisher

Secondary antibody suppliers: Abcam, Cambridge, UK; Thermo Fisher Scientific, Waltham, MA, USA.

## Data Availability

The data presented in this study are available on request from the corresponding author.

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
