# Peer review of "Endocannabinoid System Receptors at the Hip and Stifle Joints of Middle-Aged Dogs: A Novel Target for the Therapeutic Use of Cannabis sativa Extract in Canine Arthropathies"

_animals, 2023, doi:10.3390/ani13182833_

Round 1

Reviewer 1 Report

1)   Summary

The paper aims to identify the expression and distribution of cannabinoid receptor 1 (CB1), 2 (CB2) and the cannabinoid-like receptor GPR55 in the synovial cells of the hip and stifle joints of healthy dogs.

The paper presents new information on cannabinoid receptors in the healthy canine joint. The authors have previously published similar work in identifying CB1, CB2, GPR55, TRPV1 and PPARα in the equine synovial membrane, and therefore one aspect of novel findings lies in the specific identification of CB1, CB2 and GPR55 receptors in the canine synovium. In addition, the authors found that, unlike the horse, CB1 and CB2 receptors are readily found in the canine synovial membrane.

The receptor findings are novel and interesting.

2)  General concept comments.
There are areas this reviewer perceives as weaknesses in this paper:

(1)              The Western blot images chosen for Figure 1 fall very short of what is expected in a publication, including narrowly cropped images and missing molecular weight marker channels. There are no positive or negative control channels shown in the selection. The presented images reduce the credibility of this aspect of the work. The supplementary files provide better images, although the GPR55 image does not clearly show a positive and negative control.

(2)              Analysis of “cell layers” and “cell density” of the synovial membrane: The reasoning for counting "layers" or “density” is not clear, and it was not established as a goal in the introduction. Is what was described the synovial membrane intima?  Without an accompanying H+E histological staining of the tissues, removing this aspect from the publication may be best.

(3)              Animals and joints deemed "healthy", but then a major focus in the introduction and discussion on targeting inflammation.

Individual points:

Line 25 and elsewhere

Whether these dogs could be considered “normal” or “healthy” is debatable. All but one dog, #5, would be considered at least “middle-aged”; many would be considered "geriatric", which might be expected to influence the joint at the histological level. Several have been diagnosed with serious disease necessitating euthanasia. The authors have alluded to age and osteoarthritis in the introduction (lines 80-88). Advise remove reference to “healthy” and replace with phrasing such as “without overt signs of OA” (presumably, the joints were examined, and this statement is true- see comments on materials and methods). As the paper goes into a great deal of discussion on inflammation and cannabinoid receptors,  I believe an opportunity has been lost to describe these animals as they are- mostly aged and potentially having age-related changes to their joints- which would then justify the discussion on inflammation.

Lines 102-104. Regarding the aims/objectives- the authors presumably had a working hypothesis that there would be a difference in expression in CB1 and CB2 receptors, reflecting the previous findings in the horse. It may be easier to explain “why” comparisons were made regarding distribution if this hypothesis was stated clearly in the introduction.

Materials and methods:

Lines 108-109. Was each joint examined for overt signs of osteoarthritis? How were these joints determined to be “healthy”, given the disease presentations?

Lines 118-126: The tissue collection section needs more detail in this description for the reader to understand the work, without having to refer to another paper. For example:

·         What size were the dissected sections?

·         The synovial membrane was pinned "upward"; the inference therefore is that both the synovial membrane and synovial capsule was "harvested", and presumably using a scalpel- all this must be stated.

·         How were the tissues orientated during slicing, and how is this seen on the resulting slides (i.e. full depth of synovial membrane, from lumen/articular cavity to capsule on each slide), images shown with lumen at the top of the image?

·         The cryosection thickness must be provided.

·         Was any normal structural histology/histopathology (H+E) carried out?

·         What approx. size of tissue was placed in the effendorf tube? Were these tissues co-located with those taken for immunohistochemistry?

Lines 128-136

Somewhere within these lines, specifically mention the fluorochromes and emission spectra - the reader may not understand the significance of the numbers in Table 2. It may be useful to state that the same fluorochrome label was used for the three receptors allowing for quantification. State that DAPI emits blue fluorescence upon binding to AT regions of DNA.

Lines 197-199. Technically, slices are 3-dimensional (µm3), and cell counts also depend on the thickness of the slice. 

Results

Lines 273- 291: As stated previously, the purpose of this aspect of the paper is unclear. These numbers do not add meaningfully to knowledge on “the cellular composition in the hip and stifle joints of dogs”, and are not more informative than, for example, Sagiroglu (2012). This paper component possibly detracts from the more interesting findings, such as the descriptions in sections 3.4, 3.5 and 3.6.

Suggest removing the analysis of cell layers and density from the paper.

Lines 381-383 Again, questioning the term “healthy”.

Lines 381-387: Please rephase this section for clarity, stating more clearly the nature of the differences seen ie: “There was less expression of CB1R, in comparison with CB2R and GPR55-IR, in both the hip and the stifle joints of dogs (~p value).

Discussion and conclusion

Again, the discussion and, potentially, the conclusion should reflect the “aged” nature of most of the animals in this study.

Lines 455-472: A large discussion component reflects inflammation and cannabinoid receptors’ role in inflammation and as potential drug targets. However, this paper does not experimentally delve into inflammation and repeatedly refers to these animals and joints as “healthy”. As a result, this reviewer considers that this portion of the discussion should be substantially reduced, unless justified by a change to the description of these animals.

Figures and Images

As stated previously, the Western Blot image in Figure 1 is not publishable-  see Kroon et al. 2022 https://doi.org/10.1371/journal.pbio.3001783. Many readers will not download the supplementary files.

Figures 2-5. The immunohistochemistry images are well presented.

Figures 6 and 7.  Please provide units (ie IHC values, arbitrary units). Please indicate what the error bars etc. indicate (CI, SD, SEM?).

Unclear statement in Figure 7: “D) When comparing both articulation site receptor expression, there was a difference between both joints; however, the statistical analysis carried out showed that there was no significant difference between the receptor expression of the different joints.”

Please rephrase for clarity, as the statements appear contradictory.

Reference

Sagiroglu, A.O. 2012, "The light microscopic study of age-associated changes in dog synovial membrane cells", Journal of animal and veterinary advances : JAVA, vol. 11, no. 18, pp. 3369-3377.

The English is very good overall, however with some minor typographical errors, and places where there it is not clear what the authors are saying due to issues with phrasing.

Author Response

1) Summary

The paper aims to identify the expression and distribution of cannabinoid receptor 1 (CB1), 2 (CB2) and the cannabinoid-like receptor GPR55 in the synovial cells of the hip and stifle joints of healthy dogs.

The paper presents new information on cannabinoid receptors in the healthy canine joint. The authors have previously published similar work in identifying CB1, CB2, GPR55, TRPV1 and PPARα in the equine synovial membrane, and therefore one aspect of novel findings lies in the specific identification of CB1, CB2 and GPR55 receptors in the canine synovium. In addition, the authors found that, unlike the horse, CB1 and CB2 receptors are readily found in the canine synovial membrane.

The receptor findings are novel and interesting.

Response: We thank the Reviewer for his/her appreciation of the manuscript.

2)  General concept comments.

There are areas this reviewer perceives as weaknesses in this paper:

(1)              The Western blot images chosen for Figure 1 fall very short of what is expected in a publication, including narrowly cropped images and missing molecular weight marker channels. There are no positive or negative control channels shown in the selection. The presented images reduce the credibility of this aspect of the work. The supplementary files provide better images, although the GPR55 image does not clearly show a positive and negative control.

Response:

We apologize to Reviewer#1 for the low quality of the figures. We agree with the Reviewer's concern; in the original submission of the manuscript, we included (as requested by Animals), a ZIP folder with the original Western blot in which we added the reference molecular weights next to the bands recognised by the antibodies. All the figures are original without having been cropped. Negative controls were carried out for each antibody (anti-CB1, anti-CB2 and anti-GPR55) by incubating the membranes with only the secondary antibody. A statement citing the type of negative controls carried out using western blot analysis was added to the Materials and Methods section. In the first version of the manuscript, we cropped the entire image to make the bands and their specificity stand out better in order to avoid confounding elements given by the reference molecular weight bands. We would also like to emphasise that, in terms of specificity, all the antibodies we used had been widely tested using Western blot by our group and by others as referenced in:

  1. The rabbit antibody against CB1R from Byorbit (Orb10430) was used in Galiazzo et al., Cell Biol. 2018, 150, 187–205, doi:10.1007/s00418-018-1684-7 and Chiocchetti et al., Front. Vet. Sci. 2022, 9, 915896, doi:10.3389/fvets.2022.915896.
  2. The mouse antibody against CB2R from Santa Cruz (sc-293188) was used in Chiocchetti et al., Vet. Sci. 2022, 9, 915896, Chiocchetti et al., Front. Vet. Sci. 2022, 9, 915896, doi:10.3389/fvets.2022.915896.
  3. The rabbit antibody against GPR55 from Novus Biol (NB110-55498) was used in Chiocchetti, et al., Vet. Sci. 2019, 6, 313, doi:10.3389/fvets.2019.00313 and Galiazzo et al., Histochem. Cell Biol. 2018, 150, 187–205, doi:10.1007/s00418-018-1684-7

It should be noted that for the antibody against CB1R, there is a large body of literature regarding different cells and tissues, showing molecular weights ranging from ∼35 to ∼70 kDa: 

Pietrovito, L., et al. 2020. Treatment with cannabinoids as a promising approach for impairing fibroblast activation and prostate cancer progression. Int. J. Mol. Sci. 21: 787;

Rodrigues, R.S., et al. 2017. Interaction between cannabinoid type 1 and type 2 receptors in the modulation of subventricular zone and dentate gyrus neurogenesis. Front. Pharmacol. 8: 516;

Costa, L., et al. 2021. The major endocannabinoid anandamide (AEA) induces apoptosis of human granulosa cells. Prostaglandins Leukot. Essent. Fatty Acids 171: 10231.

Poddighe et al., 2018, Acute administration of beta-caryophyllene prevents endocannabinoid system activation during transient common carotid artery occlusion and reperfusion. Lipids in health and disease, 17(1), 23. doi: 10.1186/s12944-018-0661-4.

Moreover, our data showed that, as for the anti-CB2R (molecular weights ranging from ∼35 and ∼60 kDa) and anti-GPR55 (molecular weights ∼35 kDa) antibodies, the molecular weights we identified were the same as those reported not only in the literature but also in the relevant datasheets provided by the manufacturer.

Following the Reviewer’s suggestions, we replaced the original Figure 1 with a new Figure 1 in which the molecular weights of the identified bands and negative controls have been added.

(2)              Analysis of “cell layers” and “cell density” of the synovial membrane: The reasoning for counting "layers" or “density” is not clear, and it was not established as a goal in the introduction. Is what was described the synovial membrane intima?  Without an accompanying H+E histological staining of the tissues, removing this aspect from the publication may be best.

Response: We agree with Reviewer that the studies of the cell layers and cell density seem off topic and that they were not described as a goal of the study. Therefore, as suggested by the Reviewer, any details related to this topic were deleted (paragraphs 3.2 and 3.3).

(3)              Animals and joints deemed "healthy", but then a major focus in the introduction and discussion on targeting inflammation.

Response: We agree with the Reviewer and have changed “healthy” to “without overt signs of OA“  when referring to the dogs analysed in the current study.

Individual points:

Line 25 and elsewhere

Whether these dogs could be considered “normal” or “healthy” is debatable. All but one dog, #5, would be considered at least “middle-aged”; many would be considered "geriatric", which might be expected to influence the joint at the histological level. Several have been diagnosed with serious disease necessitating euthanasia. The authors have alluded to age and osteoarthritis in the introduction (lines 80-88). Advise remove reference to “healthy” and replace with phrasing such as “without overt signs of OA” (presumably, the joints were examined, and this statement is true- see comments on materials and methods).

Response: We agree with the Reviewer and have changed “healthy” to “without overt signs of OA“.

As the paper goes into a great deal of discussion on inflammation and cannabinoid receptors, I believe an opportunity has been lost to describe these animals as they are- mostly aged and potentially having age-related changes to their joints- which would then justify the discussion on inflammation.

Response: We thank the Reviewer for his/her clarifying perspective regarding our study and we agree. Changes have been made (Pag. 3, lines 126-128) regarding the description of the animals (Pag. 18, lines 606-615).

 Lines 102-104. Regarding the aims/objectives- the authors presumably had a working hypothesis that there would be a difference in expression in CB1 and CB2 receptors, reflecting the previous findings in the horse. It may be easier to explain “why” comparisons were made regarding distribution if this hypothesis was stated clearly in the introduction.

Response: The hypothesis is that the studied receptors would have different expression and cellular distribution. We added a sentence in the “introduction” (Pag 3, lines 117-120).

Materials and methods:

Lines 108-109. Was each joint examined for overt signs of osteoarthritis? How were these joints determined to be “healthy”, given the disease presentations?

Response: A sentence has been added to the M&M (Tissue collection) in which we specified that all the synovial membranes were analysed for the presence of OA (Pag. 4, lines 140-142).

 Lines 118-126: The tissue collection section needs more detail in this description for the reader to understand the work, without having to refer to another paper. For example:

  • What size were the dissected sections?

Response: The size of the tissue samples (which was the same as for the tissue sections) has been added (Pag. 4, lines 142-143).

  • The synovial membrane was pinned "upward"; the inference therefore is that both the synovial membrane and synovial capsule was "harvested", and presumably using a scalpel- all this must be stated.

Response: The requested detail has been added (Pag. 4, line 144).

  • How were the tissues orientated during slicing, and how is this seen on the resulting slides (i.e. full depth of synovial membrane, from lumen/articular cavity to capsule on each slide), images shown with lumen at the top of the image?

Response: The full thickness of the joints (synovial membrane + synovial capsule) was collected. During slicing, the full thickness of the synovial membrane was visible; the luminal lining was mostly visible for the density of DAPI labelled nuclei of the synoviocytes and for the density of the capillaries in the subintima.

To increase the comprehensibility of the reader, more information has been added to the images (intima and subintima) and the legends of the microphotographs.

  • The cryosection thickness must be provided.

Response: The thickness of the sections has been added (14µm thick) (Pag. 4, line 148).

  • Was any normal structural histology/histopathology (H+E) carried out?

Response: No H+E sections were prepared and analysed; in fact, in the cryosections, all the nuclei of the synoviocytes, as well as all the macrophages and neutrophils, were detected in the entire thickness of the synovial membrane with specific markers (IBA1 and MAC387, respectively). In addition, an endothelial marker (CD 31) was also utilised.

  • What approx. size of tissue was placed in the eppendorf tube? Were these tissues co-located with those taken for immunohistochemistry?

Response: The tissues for the immunofluorescence and Wb were absolutely not co-located together. More specifically, for immunofluorescence, the tissues must be oriented and distended with entomological pins before fixation in small containers. For Wb analysis, no fixation of the tissues is permitted/required, and the tissues must be collected as soon as possible after death and placed in liquid nitrogen (after putting them in an Eppenderf tube).

Lines 128-136

Somewhere within these lines, specifically mention the fluorochromes and emission spectra - the reader may not understand the significance of the numbers in Table 2.

Response: The required information related to the filters has been added (Pag. 6, lines 262-267).

It may be useful to state that the same fluorochrome label was used for the three receptors allowing for quantification.

Response: We agree with Reviewer and have added a sentence stating that the same fluorochrome label was used for the three receptors, allowing quantification (with the exception of the co-localisation studies between CB1R and CB2R (mouse anti-CB2R) (Pag. 6, lines 269-270).

State that DAPI emits blue fluorescence upon binding to AT regions of DNA.

Response: This information has been added, accordingly (Pag. 4, lines 162-164).

Lines 197-199. Technically, slices are 3-dimensional (µm3), and cell counts also depend on the thickness of the slice.

Response: We agree with Reviewer. However, the relative parts of the cell layers and density were removed from the manuscript.

Results

Lines 273- 291: As stated previously, the purpose of this aspect of the paper is unclear. These numbers do not add meaningfully to knowledge on “the cellular composition in the hip and stifle joints of dogs”, and are not more informative than, for example, Sagiroglu (2012). This paper component possibly detracts from the more interesting findings, such as the descriptions in sections 3.4, 3.5 and 3.6.

Suggest removing the analysis of cell layers and density from the paper.

Response: We agree with Reviewer. The two paragraphs (3.2 Analysis of cell layers of the synovial membrane) (3.3 Analysis of the cell density of the synovial membrane) have been removed.

Lines 381-383 Again, questioning the term “healthy”.

Response: The word “healthy” has been removed.

 Lines 381-387: Please rephase this section for clarity, stating more clearly the nature of the differences seen ie: “There was less expression of CB1R, in comparison with CB2R and GPR55-IR, in both the hip and the stifle joints of dogs (~p value).”

Response: The section has been rephrased, accordingly (Pag. 14, lines 443-454).

Discussion and conclusion

Again, the discussion and, potentially, the conclusion should reflect the “aged” nature of most of the animals in this study.
Response: We thank the Reviewer for this suggestion. Some sentences relating to the age of the animals considered have been added (Pag. 18, lines 606-615, line 619, line 639).

Lines 455-472: A large discussion component reflects inflammation and cannabinoid receptors’ role in inflammation and as potential drug targets. However, this paper does not experimentally delve into inflammation and repeatedly refers to these animals and joints as “healthy”. As a result, this reviewer considers that this portion of the discussion should be substantially reduced, unless justified by a change to the description of these animals.

Response:  We thank the Reviewer for his/her clarifying point of view, and, overall, we agree with changing the description of the animals putting a focus on middle/advanced-aged animals; changes have been made to the Discussion (Pag. 18, lines 606-615).

Figures and Images

As stated previously, the Western Blot image in Figure 1 is not publishable- see Kroon et al. 2022 https://doi.org/10.1371/journal.pbio.3001783.

Response: A new Figure 1 has been prepared, as written above.

Many readers will not download the supplementary files.

Response: Yes, this is true. However, at present, in the revised manuscript, there is only one supplementary figure (Figure S1), showing the co-localisation between the anti-GPR55 and the anti-CD31 antibodies.

Figures 2-5. The immunohistochemistry images are well presented.

Response: We thank the Reviewer for his/her appreciation of the Figures 2-5.

Figures 6 and 7.  Please provide units (ie IHC values, arbitrary units). Please indicate what the error bars etc. indicate (CI, SD, SEM?).

Response: We thank the Reviewer. The units of the images have been changed. The bars are mean with SD; information has been added to the subtitles.

Unclear statement in Figure 7: “D) When comparing both articulation site receptor expression, there was a difference between both joints; however, the statistical analysis carried out showed that there was no significant difference between the receptor expression of the different joints.”
Please rephrase for clarity, as the statements appear contradictory.

Response: The statement regarding Figure 6 has been corrected (we think the Reviewer was referring to Figure 6, not to Figure 7) (Pag. 15, lines 467-470).

Reference

Sagiroglu, A.O. 2012, "The light microscopic study of age-associated changes in dog synovial membrane cells", Journal of animal and veterinary advances : JAVA, vol. 11, no. 18, pp. 3369-3377.

Response: We thank the Reviewer for his/her suggestion; however, since the part of the study relating to the cell density and the thickness of the synovial membrane has been deleted, we did not use this new reference.

Reviewer 2 Report

This is a highly relevant and interesting study about cannabinoid receptors. Since current pain management is trying to adopt multimodal protocols and drugs that act specifically in certain receptors, the present findings Will be of great use in the understanding and management of OA. I just left some minor comments regarding the manuscript.

Line 13: Please, revise the Instructions for Authors to add a simple summary.

Line 20: Include the sample size and that the dogs were from different breeds.

Line 35: I consider it important to mention which endogenous cannabinoids are part of the ECS.

Line 40: I would recommend stating the name of the Cannabis gender (e.g., Cannabis sativa or Cannabis indica).

Line 48-49: If AEA and 2-AG can be found in a healthy dog’s joints, please, state so in this sentence.

Line 62: Here, the authors could mention the physiological role of changing the ion flow or the inhibition of neurotransmitter release. This article might be helpful:  https://doi.org/10.3389/fvets.2022.1050884.

Line 87: Another issue with animals suffering from OA is the difficulty in managing pain, particularly chronic pain, a consequence that will be present in most patients. By researching cannabis and the ECS, more options for multimodal analgesia protocols could be performed, and I consider this a relevant application of the present study.

Line 91 and 94: Add references to back up this information.

Line 109: I consider it relevant to include some other inclusion/exclusion criteria for the animals. For example, if the included dogs were healthy subjects, or if animals suffering from OA diseases were also included. How or why did the authors specifically select these seven dogs? Also, the age difference between animals is large (2-15 years), so it is important to mention what was the selection criteria.   

Line 266: Please, revise the Instructions for Authors to amend the style to cite Figures in the text (e.g., “Figure 1” instead of (Fig. 1”).

Line 382: Please, add the exact p-value.

Figure 6. This is a very interesting Figure. Maybe the authors could add the place where statistical significance was found.

Line 484: Use the abbreviation for “cannabinoid receptor 2”. The same applies to G protein-coupled receptor 55.

Decision: Accept with minor changes.

Author Response

This is a highly relevant and interesting study about cannabinoid receptors. Since current pain management is trying to adopt multimodal protocols and drugs that act specifically in certain receptors, the present findings Will be of great use in the understanding and management of OA. I just left some minor comments regarding the manuscript.

Response: We thank the Reviewer for his/her appreciation of the study

Line 13: Please, revise the Instructions for Authors to add a simple summary.

Response: We thank the Reviewer for this suggestion. A simple summary has been added as requested and suggested.

Line 20: Include the sample size and that the dogs were from different breeds.

Response: We thank the Reviewer. The sample size was included and the different breeds of the dogs have been stated (Pag. 1, lines 29-30).

Line 35: I consider it important to mention which endogenous cannabinoids are part of the ECS.

Response: The name of the two endocannabinoids have been added to the text (Pag. 2, lines 45-46).

Line 40: I would recommend stating the name of the Cannabis gender (e.g., Cannabis sativa or Cannabis indica).

Response: The name has been added, as suggested (Pag. 2, line 51).

Line 48-49: If AEA and 2-AG can be found in a healthy dog’s joints, please, state so in this sentence.

Response: The sentence (Pag. 2, lines 63-63) was modified, as suggested. In addition, the names of the endocannabinoids have been added, accordingly (Pag. 2, line 45-46).

Line 62: Here, the authors could mention the physiological role of changing the ion flow or the inhibition of neurotransmitter release. This article might be helpful:  https://doi.org/10.3389/fvets.2022.1050884.

Response: A sentence has been added (and a new reference: Rea et al., 2007), as suggested (Pag. 2, lines 60-62).

Line 87: Another issue with animals suffering from OA is the difficulty in managing pain, particularly chronic pain, a consequence that will be present in most patients. By researching cannabis and the ECS, more options for multimodal analgesia protocols could be performed, and I consider this a relevant application of the present study.

Response: We appreciate the clarifying point of view of the reviewer and the support of our research.

Line 91 and 94: Add references to back up this information.

Response: We thank the Reviewer for his/her observation. As suggested, a new reference has been added (Pag. 3, line 105).

Line 109: I consider it relevant to include some other inclusion/exclusion criteria for the animals. For example, if the included dogs were healthy subjects, or if animals suffering from OA diseases were also included. How or why did the authors specifically select these seven dogs? Also, the age difference between animals is large (2-15 years), so it is important to mention what was the selection criteria.   

Response: We thank the Reviewer for her/his comment. The animals were selected by the organic status of the joints, the absence of OA in the joints studied and no clinical history of OA or lameness related to joint pain. The authors have inserted a modification into the text (Pag. 3, lines 126-128) regarding the selection criteria of the animals included in the text.

Line 266: Please, revise the Instructions for Authors to amend the style to cite Figures in the text (e.g., “Figure 1” instead of (Fig. 1”).

Response: We thank the Reviewer and changed “Fig.” with “Figure” throughout the text.

Line 382: Please, add the exact p-value.

Response: The paragraph has been rephrased and the p values have been added to the text (Pag. 14, lines 443-454).

Figure 6. This is a very interesting Figure. Maybe the authors could add the place where statistical significance was found.

Response: The authors thank the Reviewer for her/his comments and appreciation. The figure has been modified accordingly and a bar with * and ** was placed on the top of the receptor bars where there is a statistically significant difference.

Line 484: Use the abbreviation for “cannabinoid receptor 2”. The same applies to G protein-coupled receptor 55.

Response: The native speaker corrected us and said that it would be more correct not to start a sentence without an acronym.

Decision: Accept with minor changes.

Reviewer 3 Report

The manuscript is interesting and well-written. Below are my comments.

Have the authors considered or observed differences between males and females?

A number of analyzed samples came from animals with mastocytoma. Have the authors observed any expression differences in these animals compared to the rest of the samples? This should be discussed in the text.

The study presents highly interesting results, but a limitation is the sample size. This should be discussed in the text.

Author Response

The manuscript is interesting and well-written. Below are my comments.

Response: We thank the Reviewer for his/her appreciation of the study

Have the authors considered or observed differences between males and females?

Response: The authors did perform statistical analysis to compare males and females, using paired and unpaired, parametric and non parametric tests. In both statistical methods p indicated no significant difference between males and females for any of the receptors at any joint. It is necessary to point out that the sample size was too small to perform such a comparison. In both tests, for p to be significant, it was set at P < 0.05 with a confidence interval of 95%.

A number of analysed samples came from animals with mastocytoma. Have the authors observed any expression differences in these animals compared to the rest of the samples? This should be discussed in the text.
Response: This is a very interesting point. However, only 2 dogs of the 7 had a mastocytoma, too small a number to carry out a valid statistical method of comparison

The study presents highly interesting results, but a limitation is the sample size. This should be discussed in the text.

Response: We agree with the Reviewer. The text has been modified accordingly to be clearer regarding the limitation of the study (small sample size) (Pag. 18, lines 606-608).

Round 2

Reviewer 1 Report

This reviewer appreciates the authors' efforts to address each review comment and is happy to accept this paper for publication.